# Microvascular Leakage as Therapeutic Target for Ischemia and Reperfusion Injury

**DOI:** 10.3390/cells12101345

**Published:** 2023-05-09

**Authors:** Jan Andreas Kloka, Benjamin Friedrichson, Petra Wülfroth, Rainer Henning, Kai Zacharowski

**Affiliations:** 1Department of Anaesthesiology, Intensive Care Medicine and Pain Therapy, University Hospital Frankfurt, Goethe University, 60590 Frankfurt, Germanybenjamin.friedrichson@kgu.de (B.F.); 2F4 Pharma, 1060 Vienna, Austria; 3Accella Advisors, 8737 Uetliburg, Switzerland

**Keywords:** reperfusion injury, capillary leak, edema, organ protection, clinical outcome

## Abstract

Reperfusion injury is a very common complication of various indicated therapies such as the re-opening of vessels in the myocardium or brain as well as reflow in hemodynamic shutdown (cardiac arrest, severe trauma, aortic cross-clamping). The treatment and prevention of reperfusion injury has therefore been a topic of immense interest in terms of mechanistic understanding, the exploration of interventions in animal models and in the clinical setting in major prospective studies. While a wealth of encouraging results has been obtained in the lab, the translation into clinical success has met with mixed outcomes at best. Considering the still very high medical need, progress continues to be urgently needed. Multi-target approaches rationally linking interference with pathophysiological pathways as well as a renewed focus on aspects of microvascular dysfunction, especially on the role of microvascular leakage, are likely to provide new insights.

## 1. Introduction

Cardiovascular diseases (CVD) continue to be the most important cause of illness and mortality worldwide. In 2020, it was estimated that 19.1 million deaths could be attributed to CVD globally; 85% of these deaths were due to acute myocardial infarction (AMI) or stroke [1]. Fortunately, an overall decline of acute CV events at a rate of 2–4%/year can be observed in industrialized countries, which can be attributed to the lowering of cholesterol levels and to a number of lifestyle factors. Case fatality has decreased at a similar rate due to continuous improvements in treatment procedures and post-treatment care [2]. Since the predominant cause of acute CV events is the occlusion of a coronary or cerebral artery, immediate revascularization by mechanical means (PCI and stenting) or thrombolysis is the treatment of choice. Patients with severe infarcts are now able to survive. However, as a consequence, an increase in post-MI heart failure is observed because of the weakened myocardium [3,4].

The sudden restoration of blood flow to the ischemic tissue can lead to further tissue damage as an unintended consequence. This phenomenon has been termed ischemia/reperfusion injury (IRI); it is now accepted to be a major contributor to the extent of the final lesion and therefore to clinical outcome [5]. Ischemia/reperfusion injury also plays an important role in poor clinical outcome in situations of multiple-organ or whole-body ischemia. Full recovery after cardiac arrest and resuscitation continues to be quite poor, and IRI contributes importantly to multi-organ malfunction [6,7,8]. In recovery from severe trauma with major bleeding and reperfusion it has been shown that IRI is an important factor [9]. Major thoracic-abdominal surgery with aortic cross-clamping may lead to severe cardiac, renal, hepatic, limb, spinal cord and pulmonary problems due to IRI occurrence [10,11]. For all these reasons, the prevention of the damage caused by reperfusion has become a target of intense research, both experimentally and clinically, in order to further improve outcome. A recent query in PubMed for “ischemia reperfusion injury” resulted in 71,153 entries, with the vast majority of papers having been published since the mid-1980s.

This review will discuss mechanistic aspects of ischemia/reperfusion injury in the heart and the CNS, but will also touch upon other organs (kidney, lung, liver) where the occurrence is well established as a cause of poor organ function following transplantation. We will also discuss the translation of these pathophysiological findings from animal studies into clinical outcomes. So far, the available data have not allowed clear treatment guidelines to be established, but trends are emerging.

A wealth of mechanistic studies has revealed important pathophysiological elements contributing to IRI, which have been reviewed extensively [5,12,13,14]. A focus of this review will be on microvascular dysfunction and capillary leakage as a major contributor to IRI as well as experimental and clinical evidence of its pathophysiological relevance.

## 2. Pathophysiology of Ischemia/Reperfusion Injury

Ischemia is defined as a lack of oxygen supply in tissues (hypoperfusion). Hypoperfusion leads to a vulnerable state of the affected tissue. Several mechanisms and mediators play a role in the response of the hypoxic cells to reperfusion. Oxygen free radicals, intracellular calcium excess, endothelial and microvascular dysfunction and altered cellular metabolism have all been demonstrated to play a major role (Figure 1). The restoration of blood flow to tissues that have been affected by the ischemia leads to the massive generation of reactive oxygen species (ROS) because of the impairment of antioxidative agents during ischemia. This oxidative stress promotes endothelial dysfunction, DNA damage, cell death via necrosis, apoptosis and ferroptosis and local inflammatory responses due to the accumulation of leukocytes [15]. Early studies concentrated on IRI in the ischemic myocardium in ST-elevated myocardial infarction (STEMI); a historical perspective of myocardial IRI has been elegantly provided by Jennings [16], who was the first to describe the phenomenon in dogs in 1960 [17]. While the focus of many studies has been on the heart, it has become clear that reperfusion injury also plays an important role in the CNS as a sequel of mechanical or thrombolytic recanalization after an ischemic stroke [18,19], and in impaired organ function following the transplantation of the kidney [20], liver [21] and lung [22]. There are a lot of similarities in mechanistic underpinnings in the different organs, as the resulting injury affects both the end-organ cell compartment as well as the respective microvasculature.

### 2.1. Myocardial IRI

In order to salvage as much ischemic myocardium as possible, it is mandatory to achieve timely reperfusion, preferably by primary percutaneous coronary intervention (PPCI). The unavoidable secondary damage caused by reperfusion has many pathophysiological components. In the cardiomyocyte compartment, edema/sarcolemmal rupture, calcium overload and subsequent hypercontraction, mitochondrial dysfunction with the opening of the mitochondrial permeability transition pore (MPTP), caused by the formation of reactive oxygen species (ROS) leading to apoptosis and ferroptosis and proteolysis by caspases and calpain, have all been well described [23]. In coronary circulation, microembolization through excessive platelet aggregation and accrued plaque debris and impaired vasomotion [24] cause no-reflow [25,26], which is observed in ~35% of all STEMI patients. No-reflow and capillary rupture and subsequent hemorrhage are highly predictive for adverse clinical outcome [27,28]. Importantly, the disruption of endothelial integrity [29] leads to the extravasation of fluid and macromolecules as well as the extravasation of leukocytes with tissue inflammation caused by the release of inflammatory cytokines [30]. The contribution of microvascular leakage and the evolution of myocardial edema to post-MI outcomes has been largely underestimated and has only recently become the focus of greater attention in the context of an integrative view of the function of the coronary microvasculature in the refused myocardium [31]. Coronary capillaries are easily compressed by interstitial edema and become hypoperfused. Cardiovascular magnetic resonance imaging studies using T2 weighted imaging for the quantification of edema have provided clear correlation of the extent of edema and hemorrhage with infarct size and clinical outcome [32,33]. The extent of myocardial edema is therefore an important prognostic criterion for STEMI outcome. The prevention of microvascular leak is an important therapeutic target in reperfusion injury [34]. The mechanisms involved in this endothelial disruption are discussed in more detail in Section 3.

The lymphatic system plays an important role for edema resolution. Therefore, the contribution to healing without adverse remodeling is receiving more and more attention [35].

### 2.2. IRI in the Brain

As in myocardial infarction, the restoration of blood flow is an important goal for the treatment of acute ischemic stroke. It has been clearly shown in experimental stroke models and in clinical studies that oxygen restoration may cause additional neuronal damage beyond that elicited by the ischemia itself. Oxidative stress, mitochondrial impairment and the activation of platelets complement and importantly contribute to reperfusion injury [19]. Importantly, there is clear evidence that the disruption of the blood-brain barrier (BBB) with the ultimate consequence of severe edema and eventual hemorrhagic transformation plays a decisive role in adverse outcome with accelerated neuronal destruction and neurological impairment [36,37].

### 2.3. IRI in Organ Transplantation

The transplantation of a donor organ is the best intervention for the treatment of patients with end stage organ failure. Ischemia/reperfusion injury is an inevitable complication in any organ transplantation due to the procedures involved in procuring, storing, transporting and implanting the donor organ, which lead to the (temporary) disturbance of oxygen supply. As there continues to be a shortage of viable donor organs, it is imperative to minimize the procedural organ damage, and the prevention of IRI is a major component of this effort in order to be able to use extended criteria organs and alleviate the shortage to a maximum extent. From a clinical standpoint, ischemia/reperfusion injury is connected to postponed graft function, acute and chronic rejection, as well as chronic issues with graft functionality.

As in myocardial and CNS IRI, multiple molecular pathways have been found to be actively involved including the activation of cell death programs, the impaired function of the endothelium, transcriptional reprogramming, as well as the activation of both the innate and adaptive immune systems. At the cellular and tissue levels, energy metabolism, mitochondria and cellular membranes are all critically involved. Microvascular dysfunction with increased vascular permeability and inflammation caused by the recruitment of polymorphonucleated cells (PMNs) has been found to play a decisive role. The underlying mechanisms of microvascular dysfunction in the kidney have been reviewed [20,38].

In preserved and transplanted lungs, perivascular and alveolar edema were found to be caused by morphologic alterations of ECs resulting in monolayer discontinuities followed by cell detachment. These alterations were present after the storage period and became more marked with the restoration of blood flow on revascularization [39].

In liver transplantation, IRI is initiated by damage to hepatic sinusoidal endothelial cells and the disruption of microcirculation during ex vivo preservation, which is exacerbated by reperfusion during the implantation surgery. It has been clearly demonstrated that the dysfunction of hepatic microcirculation is strongly correlated with early allograft dysfunction [40].

### 2.4. IRI Following Global Ischemia after Cardiac Arrest

Out of hospital cardiac arrest affects >300,000 patients annually in the US alone. Despite many improvements in treatment, especially in the rate of spontaneous circulation, the prognosis continues to be dire with in-hospital mortality exceeding 50% [1]. Ischemia/reperfusion injury is a frequent consequence of the successful resuscitation of patients suffering from a spontaneous cardiac arrest. The ensemble of pathophysiological effects is referred to as post-cardiac arrest syndrome (PCAS). These complications affect all organs, since the spontaneous restoration of blood flow affects all organs suffering from oxygen deprivation during the arrest. The current status of epidemiology, pathophysiology, treatment and prognostication has been reviewed in an ILCOR Consensus Statement and other summary reports [41,42]. It has been demonstrated that therapeutic hypothermia in particular is a valid strategy to improve outcome [43]. Therapeutic strategies employed for the improvement of the clinical outcome of patients with acute myocardial infarction may also benefit patients with PCAS. For instance, post-conditioning maneuvers have demonstrated benefits in a porcine model [44].

## 3. Microvascular Dysfunction and Capillary Leakage in IRI

A monolayer of endothelial cells covers the luminal surface of all blood vessels in the human body, which covers a surface area of greater than 1000 m^2^ [45]. It plays an important role in a significant number of physiological functions, including vascular tone, permeability for fluid, macromolecules and cells between the vascular lumen and tissue, as well as coagulation and angiogenesis. The disturbance of the endothelial function contributes significantly to many cardiovascular diseases [46,47].

Since the barrier function of the endothelial layer and its disruption are critically involved in IRI, it is important to understand the molecular constituents of the endothelial barrier and their regulation. This allows the identification of potential therapeutic targets that prevent the disruption of endothelial junctions and thereby reduce or prevent IRI [48].

The junction between neighboring endothelial cells contains several macromolecular complexes that contribute to the limited exchange of fluid, molecules and cells during normal hemostatic conditions.

The central structural and functional component of the adherens junction is vascular endothelial (VE)-cadherin, which is uniquely expressed in endothelial cells. It forms homophilic interactions between juxtaposed cells. It forms multicomponent complexes with several cytoplasmatic proteins, a very important one being with p120 catenin, which stabilizes VE-cadherin in the plasma membrane. α, β and γ-catenins anchor VE-cadherin to the actin cytoskeleton [49]. The stability of the intercellular VE-cadherin complexes is critically dependent on the activation state of the actomyosin skeleton of the endothelia cell. As demonstrated by Yang et al., VE-cadherin levels are reduced in the infarcted myocardium, with the lowest levels found in no-reflow sections [50]. Under steady state conditions, the actomyosin bundles form thick cortical bands that stabilize the adherens junction. Factors that increase permeability induce actin remodeling and the formation of radial stress fibers, which promote the re-localization of VE-cadherin complexes to focal junctions and facilitate their disruption, leading to increased permeability (Figure 2A).

Several lines of evidence point to the small GTPase of the Rho family as critical nodes in the regulation of endothelial contractile states. More specifically, the balance of the activation of RhoA and Rac1 is a main driver of permeability [51]. Even though this interplay is complex depending on the upstream stimulus, in most circumstances the activation of RhoA leads to stress fiber formation and actomyosin contraction, whereas Rac1 activation is correlated with the assembly of junctions between ECs and the stabilization of cortical actin bundles [52].

Another factor that drives the stability of the EC adherens junction is the phosphorylation state of VE-cadherin. Phosphorylation at tyrosines 658 and 731 destabilizes binding to p120 and β-catenin, and phosphorylation at tyrosine 685 and serine 665 induces VE-cadherin internalization and barrier disruption [53]. The tyrosine phosphorylation of VE-cadherin and therefore barrier stability is regulated by vascular endothelial protein tyrosine phosphatase (VE-PTP) (Figure 2B) [54].

Very important for the regulation of endothelial barrier function is the angiopoietin/TIE2 pathway. An important role of this system has been clearly demonstrated [55]. The most relevant members of the angiopoietin protein family are angiopoietin 1 (Ang1) and angiopoietin 2 (Ang2). Angiopoietin 1 is the natural ligand of the TIE2 receptor, which importantly contributes to maintaining the integrity of the microvascular endothelial barrier [56]. Angiopoietin 1 has been found to protect the heart against IRI by regulating VE-cadherin phosphorylation via facilitated binding between SH2 domain-containing phosphatase (SHP2) or receptor protein tyrosine phosphatase µ (PTPµ) [57]. Furthermore, it interacts with the integrin-β1/ERK/caspase-9 pathway in cardiomyoctes [57]. In a mouse IRI model, treatment with Ang1 improved myocardial function [57]. On the contrary, Ang2 is a significant biomarker of myocardial infarction [58]. It has been shown to act as an antagonist to Ang1 at the TIE2 receptor [56] and exacerbates cardiac hypoxia and inflammation after myocardial ischemia by promoting pericyte detachment, vascular leakage, increased adhesion molecular expression, the degradation of the glycocalyx and extracellular matrix and enhanced neutrophil infiltration in the infarct border area [59]. The overall activity of the Ang1/Ang2/TIE2 system is in addition rheostatically controlled by the dephosphorylating activity of VE-PTP [60,61].

Another important factor influencing endothelial barrier integrity is sphingosine 1 phosphate (S1P) via interaction with its type 1 receptor (S1P1) in endothelial cells [62]. Sphingosine 1 phosphate is a lysosphingolipid found in the HDL fraction in blood and also in red blood cells. Together with Ang1, S1P was found to be sufficient to promote endothelial barrier function. Sphingosine 1 phosphate sustains the endothelial barrier through increased homotypic VE-cadherin binding and the transportation of claudin 5 and other junctional proteins to the cell periphery [63]. It was also found to reduce IRI by promoting the phosphorylation of the gap junction protein connexin43 [64]. A positive effect of S1P was also identified in cerebral IRI, since endothelial signaling counteracts infarct expansion in a mouse model [65].

An interesting protein, which has been shown to protect the endothelial barrier, is the heat shock protein HSPA12B, the expression of which is restricted to the endothelial cell. In a model of temporary coronary occlusion in mice, HSPA12B was shown to reduce infarct size by the preservation of barrier integrity. This was achieved by attenuating the IRI-induced reduction in the expression of the tight junction protein ZO-1 and the activation of the PI3K/Akt/mTOR pathway in ECs [66].

Several lines of evidence point to microvascular leakage as an important contributor to IRI in myocardial infarction [34,67], stroke [68] and acute kidney injury [69], with the formation of edema as a pathophysiological hallmark. The prevention of excessive microvascular permeability is therefore a very valid target for pharmaceutical intervention.

## 4. Therapeutic Approaches

Pathophysiological studies identified a very significant number of targets, which could be addressed for the prevention or treatment of IRI. Many studies explore the beneficial influence of the conditioning of the ischemic tissue on physiological pathways (pre-, post- and remote conditioning) [70] and on influencing salvage pathways (RISK [71] and SAFE [72]) and mitochondrial stability [73] under reperfusion conditions. These approaches have been extensively reviewed elsewhere [74].

### 4.1. Experimental

It has been convincingly demonstrated that myocardial [63] and cerebral edema [75] constitute a major factor determining the extent of reperfusion injury. Therefore, the pharmacological prevention of increased microvascular endothelial permeability becomes a valid approach to prevent and treat IRI. A significant number of studies pursued this approach [76]. A highly interesting molecule is FX06, which constitutes the 28-mer peptide Bβ15-42. This peptide is constitutively liberated by the plasmin degradation of cross-linked fibrin [77,78,79]. It has been shown to reduce or prevent IRI in models of rodent and porcine myocardial ischemia and reperfusion, septic and hemorrhagic shock in rodents and pigs, resuscitation after cardiac arrest, in acute kidney injury and in kidney [80], lung [81], liver [82] and heart transplantation [83]. In acute transient LAD ligation in rats with 2 h reperfusion, infarct size was reduced by 50% [84], which was found to be comparable to ischemic preconditioning and superior to a CD18 MAb and the C5a antibody pexelizumab [85]. Scar formation after 30 days of reperfusion was also significantly reduced. In acute and chronic coronary ligation in pigs, this effect could be reproduced [86]. In acute and sub-chronic porcine models of global ischemia/reperfusion and hemorrhagic shock, FX06 was not only cardio-protective, but also reduced pulmonary, liver and small intestine damage and improved neurological recovery [87,88]. In rodent models of cardiopulmonary resuscitation, FX06 was demonstrated to improve survival and neurocognitive recovery; these effects could be attributed to the amelioration of capillary leakage [89]. The administration of FX06 at different time points in a mouse model of acute kidney injury elicited by the clamping of the renal artery for 30 min followed by reperfusion significantly reduced endothelial activation and lowered the tissue infiltration of neutrophils as well as tissue levels of neutrophil gelatinase-associated lipocalin (NGAL) [90,91,92]. Ischemia/reperfusion injury was significantly attenuated in a rat cardiac transplant model when FX06 was added to the cardioplegic solution. Treated animals showed significantly less myocardial necrosis and decreased values of cardiac troponin, a reduced number of infiltrating leukocytes and superior cardiac output [83]. In a murine allograft renal transplant model, FX06 improved the survival of recipients during the 28-day follow-up (60% versus 10%). Treatment decreased leukocyte infiltration, the expression of endothelial adhesion molecules and proinflammatory cytokines. Treatment significantly attenuated allogenic T cell activation and reduced graft rejection. Moreover, FX06 significantly reduced tubular epithelial damage and apoptosis [80]. Finally, in models of bacterial sepsis [93] and viral infections (Lassa [94], mouse thrombocytopenia syndrome virus [95] and Dengue [96]), which are known to be accompanied by severe microvascular leakage, FX06 was demonstrated to improve survival and hemodynamic changes; these effects could be attributed to the stabilization of the endothelial barrier.

The mechanism of action by which FX06 stabilizes the endothelial barrier has been thoroughly studied by several groups. FX06 competes with the binding of the fibrin E1 fragment to VE-cadherin and thereby prevents the dissociation of intercellular VE-cadherin complexes [84]. It was also shown that the transmigration of leukocytes was prevented in this way [84]. The compound rebalances the activation state of RhoA/Rac1 (reducing RhoA and increasing Rac 1 activity) in favor of endothelial cell junction stability by counteracting stress fiber formation via the prevention of myosin light chain phosphorylation. This is mediated by the dissociation of the src-family kinase Fyn, which dissociates from VE-cadherin at adherens junctions upon the exposure of endothelial cells to the peptide. Following that, Fyn binds with p190RhoGAP and hinders the activation of RhoA. Furthermore, the prevention of endothelial cell contraction also involves FAK, as the peptide causes the diffuse distribution of FAK in the cytosol instead of the localization of FAK at the tip of stress fibers [96]. The outside-in signal that causes the Fyn dissociation from VE-cadherin may be mediated by the binding of FX06 to the VLDL receptor [97,98].

CU06-1004, a pseudo-sugar derivative of cholesterol, is a novel compound under investigation for multiple disease entities characterized by increased microvascular leakage. It has been shown to reduce infarct size in models of rodent myocardial IRI by enhancing vascular integrity, improving cardiac remodeling and suppressing edema and inflammation [99]. It reduces cerebral edema and CNS IRI by suppressing blood-brain barrier disruption and cerebral inflammation [100]. The amelioration of astrocyte end-feet swelling contributes to this beneficial effect [101]. Via the modulation of colonic vessel dysfunction, experimental ulcerative colitis can be alleviated [102]. In a mouse model of diabetic retinopathy, retinal vascular leakage is prevented [103,104]. CU06-1004 induces vascular normalization and improves immunotherapy by modulating the tumor microenvironment via cytotoxic T cells [105]. The compound has been shown to achieve its protective activity via the cAMP/Rac/cortactin pathway, but the exact mechanism of signal transduction remains to be fully elucidated [106].

The angiopoietin/TIE2 pathway has also been exploited to reduce and treat ischemia/reperfusion injury. Angiopoietin-like protein 4 (ANGPTL4), a secreted 55-kDa protein that is processed into 20-kDa and 35-kDa forms, was shown to be elevated following MI in mice and humans. AGPTL4-deficient mice display (1) increased size of myocardial infarcts, (2) increased no-reflow, (3) decreased vascular integrity through Src-dependent dissociation of the VEGFR2/VE-cadherin complex and the subsequent destabilization of endothelial adherens junctions and (4) increased hemorrhage and inflammation [107]. Recombinant ANGPTL4 reduces no-reflow, hemorrhage and infarct size after myocardial infarction in mice and rabbits [102]. It also attenuates intestinal barrier structure and function after ischemia/reperfusion [108]. Interestingly, the traditional Chinese medicine Tongxinluo is also active against myocardial IRI by stimulating the expression of ANGPTL4 [109]. Another approach to target the angiopoietin/TIE2 system was demonstrated by the application of vasculotide, a short synthetic peptide (HHHRHSF), which binds with high affinity to TIE2 and activates the receptor similar to Ang1. Vasculotide prevents IRI-induced renal vascular leakage and congestion, improves renal function and recovery and significantly attenuates mortality in an AKI model [110]. An angiopoietin-1 variant engineered for higher potency and stability (COMP-Ang1) has been demonstrated to improve endothelial dysfunction in various models. It preserved renal tubulary capillaries, reduced tubular injury and decreased interstitial fibrosis in a rat acute kidney injury model [111]. It also prevented IRI and reduced allograft rejection when donor hearts were pretreated with the compound [112].

The specific small molecule VE-PTP inhibitor razuprotafib (AKB-9778) acts as an indirect activator of TIE2 by alleviating negative receptor regulation. It has been shown to stabilize microvascular integrity and is under investigation in diabetic macular edema [60]. Its application in myocardial and cerebral IRI has not been attempted yet.

The barrier-stabilizing effect of S1P1 has been the subject of several studies. For instance, fingolimod (FTY720) has been shown to exert neuroprotective effects and BBB stabilization in a rat stroke model [113]. SAR247799 is a G-protein selective S1P1 agonist with endothelial protective effects in rats and pigs at doses that do not desensitize S1P1. Preclinical studies with SAR247799 showed renal and coronary vasculature protective effects when administered prior to an insult of ischemia/reperfusion-induced endothelial injury [114]. The product is now in phase 2 clinical trials (see Section 4.2).

The PI3K/Akt/HIF-1α pathway, which is also a target for the EC heat shock protein HSPA12B, interestingly is involved in the amelioration of hypoxia/reperfusion-induced endothelial cell dysfunction by the opioid analog remifentanil [115].

Human relaxin-2 is a hormone structurally related to insulin with pleiotropic functions. It has been shown to reduce the area of no-reflow primarily by the reduction of microvascular leakage via the stabilization of VE-cadherin in the endothelial adherens junction [116]. A family 1 receptor selective relaxin 2 analog (B7-33) was shown to reduce infarct size in a mouse model. This effect is linked to the signaling of the MAP kinase pathway [117].

Imatinib (Gleevec^®^) is a prototypical multi-kinase inhibitor. It has been shown to reduce capillary leakage in multiple organ systems. Its main target Abl2 has been implicated in the loss of microvascular integrity. Using Evans-blue staining, imatinib was shown to reduce the formation of edema and hemorrhage in a rat cardiac infarct model [118]. It may be worthwhile to investigate other kinase inhibitors for their influence on capillary leakage since multiple kinase pathways are involved in the regulation of endothelial contractility.

Microvascular leakage also plays an important role in reperfusion injury subsequent to ischemic stroke. Tetrahydrocurcumin (THC), a major bioactive constituent of turmeric, has been demonstrated to abrogate brain edema and microvascular leakage in brain parenchyma, most likely through an epigenetic mechanism [119].

### 4.2. Clinical

Myriads of experimental studies over the last few decades have shown evidence of infarct size reduction and cardio-protection using a plethora of approaches, either by various conditioning strategies or by pharmaceutical interventions. Many receptors and signaling pathways that have been shown to potentially participate in myocardial IRI have been used as targets. However, the translation of these encouraging results into clinical outcomes has met with significant frustration. The results are equivocal at best in large, controlled trials, which used hard clinical endpoints such as cardiac and overall mortality and/or combined clinical outcomes (major adverse cardiovascular events, MACE). An exception is coronary ischemic preconditioning in cardiac surgery, where a clinical benefit on perioperative outcomes has been clearly demonstrated in a systematic review [120]. This was corroborated in a prospective trial in 329 cardiac surgery patients, where a long-term benefit was observed. All-cause mortality was assessed over 1.54 (SD 1.22) years and was lower with remote ischemic preconditioning than without [121].

The concept of post-conditioning has received a lot of experimental and clinical attention. It is specifically attractive, since applying post-conditional maneuvers is independent of the onset of ischemia and therefore conceptionally has broad utility. Post-conditioning is achieved by multiple short periods of coronary occlusion subsequent to the opening of the culprit coronary artery. The cardioprotective effects of post-conditioning were first described in a model of coronary occlusion/reperfusion in dogs [122]. It has subsequently been reproduced in other species such as rats [123] and rabbits [124]. Evidence points towards interference with multiple triggers, effectors and pathways leading to the inhibition of the opening of the PTP and delaying of cell alkalinization, reducing calcium accumulation and oxidative damage as well as the attenuation of endothelial dysfunction [125,126].

A number of translational clinical trials that applied post-conditioning in the setting of PCI for STEMI were initiated with somewhat conflicting results. Whereas some studies showed evidence of a reduction in infarct size [127,128,129,130], at least one study had a negative outcome [130]. In a recent large trial in STEMI patients undergoing PCI (DANAMI3-iPOST), post-conditioning had no effect on a combination of all-cause mortality and hospitalization for heart failure [131]. A follow-up trial (iPOST2; NCT03787745) is currently ongoing that does not allow thrombectomy. Interestingly, it can be demonstrated that postconditioning is able to reduce the formation of myocardial edema [132]. It is, however, not clear if this is a primary effect of the post-conditioning maneuver or secondary to the reduction of infarct size.

In another large trial (LIPSIA CONDITIONING), post-conditioning was combined with intrahospital remote preconditioning in patients with STEMI and PCI treatment. While there was a statistically significant improvement of the myocardial salvage index in the combined treatment group, post-conditioning alone had no effect. However, on long-term follow up (median of 3.6 years) in the combined treatment group, major cardiac events were reduced and this was driven by a reduction in new congestive heart failure [133].

Taking the available evidence together, it remains to be demonstrated that post-conditioning maneuvers have a place in clinical practice in IRI prevention and, most importantly, which patient group will benefit most from it.

The overall clinical situation is quite disappointing, as smaller trials in well-defined populations gave hope for successful translation into improving longer-term clinical outcomes. Table 1 provides a summary of the most important trials, which all failed. A very detailed analysis of clinical trials performed so far has been discussed by Heusch and Rassaf [134].

Several authors have reflected on the reasons for this poor translation of promising experimental results from animal studies [5,13,23,134,145,149,150,151,152,153,154,155,156,157]. In these detailed analyses, some consensus has been achieved. Clearly, there are major differences in the setting of animal studies vs. broad clinical application in patients. The pathophysiological mechanisms, which in their totality determine the final outcomes of myocardial infarction and reperfusion intervention, form a complex network. In order to evaluate the contribution of a presumed target or pathway, however, they are usually studied in isolation. This reductionist approach, even though required to provide robust data, which by nature neglects the interactions with other pathways, does not usually take these interactions into account. Furthermore, experimental approaches are usually performed in young healthy animals, whereas patients usually have a long history of atherosclerotic lesions, which eventually precipitate the coronary obstruction.

There is ample clinical evidence that the efficacy of cardioprotective treatment modalities is further limited in the presence of comorbidities (hyperglycemia, hypertension, hyperlipidemia). Advanced patient age also has negative effects on their efficacy [158]. These comorbidities affect changes in signaling pathways, which may have an influence on the development of IRI itself and on the response to therapeutic interventions. Diabetic patients have an increased susceptibility to IRI and lose response to ischemic conditioning maneuvers. This is likely due to increased baseline oxidative stress and an impairment of pathways protecting the myocardium from cell death [159]. Hypertension has been demonstrated to negatively impact cardioprotection. For instance, hearts from spontaneously hypertensive rats develop larger infarcts after coronary occlusion and are less reactive to postconditioning [160,161]. This is also observed in aged animals; this has been linked to changes in gene and protein expression, signal transaction cascades and deterioration in mitochondrial function [162]. Even though the influence of female hormones on susceptibility to ischemic heart disease is well known, the existence of a gender difference in the IRI response is unclear and requires further investigation [163]. In order to improve the success of new development products for cardioprotection, appropriate animal models must be employed that include observations on these confounding influences [164]. Moreover, patient selection in clinical trials is critical to avoid the dilution of clinical efficacy signals by confounding factors [134,149]. In addition, patients usually have several co-medications (e.g., antihypertensives, statins and anticoagulants) [165,166,167], which may have confounding effects, as do anesthetics that are used during CABG, for which a conditioning response has been proposed [168,169]. Most clinical trials have been performed in academic centers, which provide highest quality care. For this reason, the post-intervention event rate in clinical trials is significantly lower than in a broader population. These improvements in standard care make it much harder to detect a protective contribution of an intervention and require very large clinical trials.

Most large studies have concentrated on the protection of cardiac myocytes from the consequences of ischemia and reperfusion. In smaller proof-of-concept trials, infarct size fairly early after reperfusion was therefore used as the primary endpoint. However, the long-term correlation with post-infarct remodeling is far from clear; this may have an important impact on the longer-term outcome. Finally, the contribution of microvascular impairment in the IRI processes has received a lot less attention, and some authors suggest putting more emphasis on microcirculation as a target for cardio-protection [25,170,171,172,173,174]. Microvascular obstruction and capillary compression by myocardial edema caused by the loss of endothelial function, which result in no-reflow, have been clearly correlated with infarct size and clinical outcome [173].

The prevention of microvascular leakage has rarely been considered clinically for the reduction of IRI. The translation of the preclinical effects in leak reduction achieved with multiple experimental drugs should become a priority. It will be of great interest to investigate those compounds that have been demonstrated to stabilize endothelial integrity. A few clinical trials with such agents have been performed or are underway. The fibrin-derived peptide FX06 has been investigated in the F.I.R.E. trial in 234 patients undergoing PCI for acute STEMI [175]. Patients were stratified for door-to-balloon time and the presence/absence of collaterals. Infarct size was measured by MRI at 5–7 days and at 4 months. While there was no significant difference to placebo in infarct size as measured by total late enhancement area, some interesting observations were made, which may warrant future evaluation. The necrotic core zone, which determines the area of non-viable myocardium, was significantly reduced, as was the amount of microvascular obstruction (MVO). In patients presenting early (<3 hours time-to-reperfusion), compared to placebo there was a highly significant reduction after 4 months both in necrotic core zone (0.3% vs. 2.4%, *p* = 0.032) and in total infarct size (8.0% vs. 16.0%; *p* = 0.032), suggesting a longer-term treatment effect in these patients. Moreover, patients with collaterals have smaller infarcts after treatment with FX06 (7.3% vs. 15.2%; *p* = 0.043) [176]. Since the loss of endothelial barrier function has pathophysiological relevance in many diseases, FX06 has shown anecdotal beneficial effects in patients with Ebola infection [177] and COVID-19-induced respiratory distress syndrome [178]. This latter application is now under systematic investigation in the prospective randomized IXION trial [179].

Safety and pharmacokinetics have been investigated for the endothelial barrier stabilizer CU06-1004 [NCT04795037]. This product is now in a phase 2a clinical trial in diabetic macular edema [NCT05573100]. According to the website of the company Curacle (www.curacle.com, accessed on 5 February 2023), a clinical development adjunct to PCI in myocardial infarction is planned.

The G-protein biased sphingosine-1 phosphate receptor-1 (S1P1) agonist SAR247798 has been investigated for safety and PK [180]. A study with the measurement of improvement of endothelial function in diabetic patients has also been completed [NCT03462017]; the results have not yet been published.

The company Vasomune (www.vasomune.com, accessed on 5 February 2023) develops a PEGylated version of the Tie2 agonistic peptide vasculotide (AV-001) in acute respiratory distress caused by bacterial or viral infection [NCT05123755] and in COVID-19-related respiratory distress, which is thought to be related to endothelial barrier impairment [NCT04737486].

None of these promising approaches have been advanced to large randomized controlled trials. The results from such studies will be eagerly awaited and may provide new avenues for treatment.

## 5. Conclusions and Outlook

Reperfusion therapy is firmly established as the best therapeutic modality to salvage myocardium in patients with acute STEMI as well as in patients with cerebral ischemic stroke. Other clinical scenarios, such as reflow in hemodynamic shutdown following cardiac arrest, severe bleeding and hemorrhagic shock or aortic cross-clamping, are often not recognized in this matter, although outcome is often severely compromised. It is now a common understanding in the field that reperfusion may lead to additional tissue damage beyond the ischemia itself. This effect has been aptly called ischemia/reperfusion injury (IRI). It has been shown to also play a pathophysiological role in other organ systems, e.g., in kidney, lung and liver injury following transplantation and in respiratory distress in bacterial and viral infections. The prevention and treatment of the deleterious effects of IRI have therefore become a target of intense research over the last few decades [126]. While many encouraging results have been obtained in experimental studies, the translation into improvements in clinical outcomes has been notoriously difficult. This has even led to questions over whether a clinical implementation will be possible at all [134].

However, we believe that this would be a premature conclusion and the need for effective tissue-protecting strategies continues to be pressing. There is hope that a more comprehensive characterization of candidate treatments will reveal interventions with a higher probability of success. This approach was recently summarized in several commandments [181]:-Before being applied in a clinical setting, the effectiveness of the cardioprotective intervention must be verified in several models, preferably including large animals and comorbidities.-The data should be reproducible between different study centers in multi-center trials.-Endpoints in the latter stages of preclinical studies should mirror the clinical endpoints that will have an impact on medical practices, such as mortality.-Pre-clinical translational studies should start to reflect the background of current medical therapy for STEMI patients.

The majority of pivotal large-scale clinical trials have concentrated on the protection of cardiomyocytes from reperfusion injury. The protection of the microvasculature has comparably received a lot less attention despite ample pathophysiological evidence of the major role of microvascular leakage. The promising experimental results obtained with a diverse group of interventions that prevent microvascular leak wait to be translated into clinical applications beyond small proof of concept trials. A more detailed clinical investigation of strategies that target the integrity of the microvascular endothelium can certainly be expected to provide new avenues for tissue protection.

Finally, it will be a worthwhile endeavor to have a more integrative systems medicine-based approach, which takes the complex interactions of multiple pathways involved in IRI into account. Such multitarget approaches, which look at the myocardium and the coronary microvasculature, have already been proposed [182,183] and may ultimately provide solutions to this difficult problem. Taking all the learnings from 40 years of cardio-protection studies into account will eventually provide robust approaches for organ protection in the setting of ischemia/reperfusion injury.

## Figures and Tables

**Figure 1 cells-12-01345-f001:**
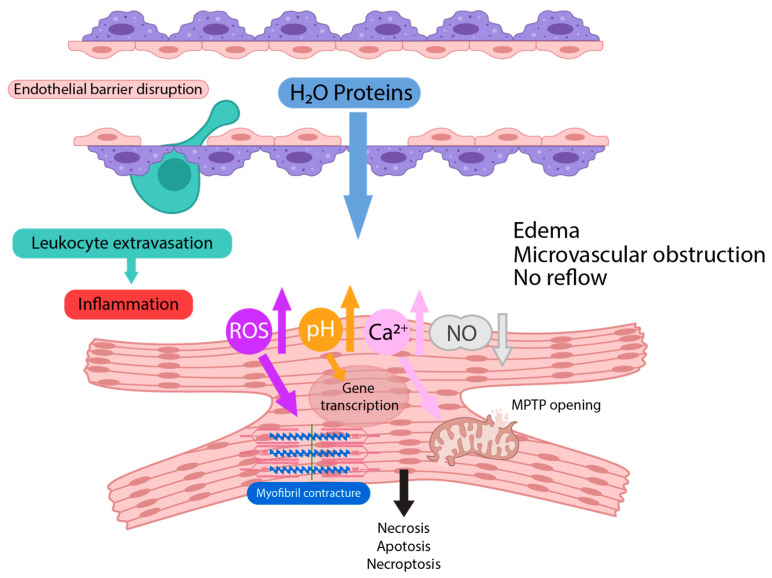
Ischemia/reperfusion injury affects the myocardium and the microvasculature. Restoration of blood flow leads to production of reactive oxygen species, normalization of pH, release of calcium and reduced availability of nitric oxide in the cardiomyocyte. These biochemical changes lead to myofibril contracture and MPTP opening in mitochondria, and eventually to necrosis, apoptosis or necroptosis. Endothelial barrier disruption leads to extravasation of water and proteins from the coronary vessels and edema, capillary compression and microvascular obstruction. Tissue inflammation is a consequence of leukocyte diapedesis.

**Figure 2 cells-12-01345-f002:**
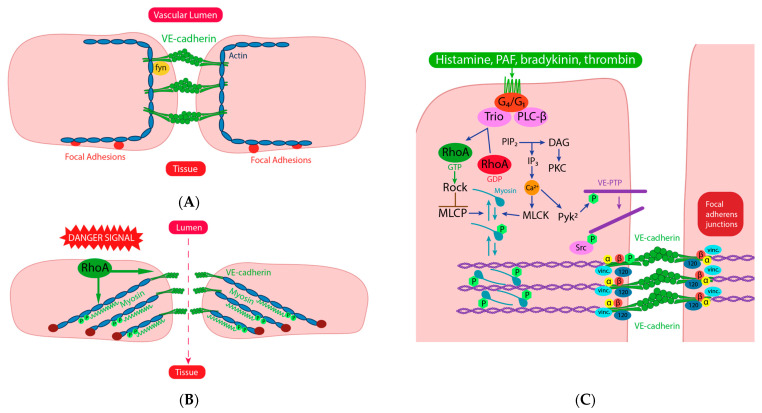
(**A**) under resting condition, the endothelial barrier is stabilized by attachment of VE-cadherin to thick cortical bands of actin fibers. (**B**) upon stimulation with danger signals, the actomyosin cytoskeleton is re-arranged with the formation of transcellular stress fibers. The contraction of myosin causes a disruption of the homotypic complexes of VE-cadherin and an opening of the adherens junction. (**C**) Rho A and Rac 1 play a central role in this process. Upon GPCR activation by various ligands, RhoA is transformed into its GTP binding form, which can activate Rho-associated kinase (ROCK), which phosphorylates myosin light chain leading to actomyosin contraction. The RhoA activity is functionally opposed by Rac1 (modified from [48]).

**Table 1 cells-12-01345-t001:** Major negative clinical trials designed for demonstration of clinical benefit of prevention/treatment of myocardial ischemia/reperfusion injury in heart surgery and PCI for STEMI.

Study Name	Type of Intervention	Number of Patients	Primary Endpoint And Outcome	Reference
ERICCA	Remote ischemic preconditioning in coronary artery bypass graft surgery	1612	MACCE Negative	[135]
RIPHeart	Remote ischemic preconditioning for heart surgery	1403	MACCE Negative	[136]
CONDI 2- ERIC-PPCI	Ischemic conditioning in STEMI patients undergoing PCI	5401	Combined cardiac death of HF hospitalization 12 months Negative Combined cardiac death of HF hospitalization 30 days Negative	[137]
DANAMI-3- iPOST	Ischemic Postconditioning During PCI for Patients With STEMI	1234	MACE Negative	[131]
CYCLE	Intravenous Bolus of Cyclosporin A before PCI in patients with STEMI	410	6-month composite (all-cause mortality, HF, cardiogenic shock) Negative, (more events in CsA group)	[138]
CIRCUS	Intravenous Bolus of Cyclosporin A before PCI in patients with STEMI	970	Composite of death or rehospitalization or worsening of HF, adverse LV remodeling at 1 year Negative	[139]
MITOCARE	Intravenous bolus of MPTP blocker TRO40303 before PCI in patients with STEMI	168	Infarct size (CK, TnI) Negative (CK AUC_0–72 h_) TnI AUC_0–72 h_)	[140]
AMISTAD	Adenosine infusion during and after PCI for patients with STEMI	236	Infarct size by SPECT imaging at 5–12 days IS reduction by 31%	[141]
AMISTAD-II	Adenosine infusion during and after PCI for patients with STEMI	2118	MACE at 6 months Negative	[142]
APEX-AMI	Bolus injection of anti-C5 antibody pexelizumab prior to PCI in patients with STEMI	5745	30-day mortality Negative MACE at 90 days Negative	[143]
PROTECTION-AMI	Infusion of delta-PKC inhibitor delcasertib during and after PCI in patients with STEMI	1010	Infarct size (CK-MB AUC) Negative (no difference in all dose groups) No difference in composite of death, HF and serious ventricular arrhythmias at 3 months	[144,145]
AIDA STEMI	Infusion of GpIIb/IIIa MAb abciximab in patients with STEMI undergoing PCI	2065	MACE at 90 days Negative	[146]
HEBE III	Single dose of erythropoietin within 3 hrs after PCI in patients with STEMI	529	Mean LVEF after 6 weeks Negative	[147,148]

## Data Availability

No new data were created or analyzed in this study. Data sharing is not applicable to this article.

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
