# Peer review of "Microvascular Leakage as Therapeutic Target for Ischemia and Reperfusion Injury"

_cells, 2023, doi:10.3390/cells12101345_

Round 1
Reviewer 1 Report
Kloka et al. comprehensively describe the importance of endothelial barrier integrity for organ function and the pathological consequences of barrier dysfunction, such as in ischemia reperfusion injury (IRI). The authors also review the IRI-relevant pathomechanisms on the molecular level with a focus on the critical role for VE-cadherin (CD144) and its signal transduction as well as on the angiopoietin/TIE-2 pathway. Along this line, Kloka et al. summarise what is currently known about experimental approaches to maintain or restore integrity of the vascular wall. They review compounds, such as the fibrin-derivate FX06 and drugs interfering with the angiopoetin/TIE-2 pathway as well as their mechanism(s) of action. However, with regard to clinical data available so far many of the experimental approaches did not lead to a broader use in human subjects, mainly due to the fact that the pre-clinical settings did not take comorbidities into account. The authors conclude to investigate multifactorial approaches based on a systems medicine level.
Author Response
Point-by-Point response to the comments of the editor and the reviewers
Cells – SubmissionID: cells-2325034
Microvascular Leakage as Therapeutic Target for Ischemia and Reperfusion Injury
We would like to thank the reviewers for their thorough review of our manuscript and their valuable and helpful comments. We have revised parts of the manuscript and integrated new detailed information for the reader on the basis of the excellent comments.
The reviewers' comments partly overlap thematically. We have tried to incorporate all ideas and comments. Due to the narrative structure, we have had to place some points in several places. To make it easier to find them, you will find the changes highlighted in the manuscript. We are convinced that this has significantly improved the quality of the manuscript. Please find below the point-to-point revision.
Comment 1:
Kloka et al. comprehensively describe the importance of endothelial barrier integrity for organ function and the pathological consequences of barrier dysfunction, such as in ischemia reperfusion injury (IRI). The authors also review the IRI-relevant pathomechanisms on the molecular level with a focus on the critical role for VE-cadherin (CD144) and its signal transduction as well as on the angiopoietin/TIE-2 pathway. Along this line, Kloka et al. summarise what is currently known about experimental approaches to maintain or restore integrity of the vascular wall. They review compounds, such as the fibrin-derivate FX06 and drugs interfering with the angiopoetin/TIE-2 pathway as well as their mechanism(s) of action. However, with regard to clinical data available so far many of the experimental approaches did not lead to a broader use in human subjects, mainly due to the fact that the pre-clinical settings did not take comorbidities into account. The authors conclude to investigate multifactorial approaches based on a systems medicine level.
Response 1: Thank you for your feedback, we enjoyed reading it. Your kind words indicate that you are satisfied with our review. If you have any suggestions for improvement, please let us know.
Reviewer 2 Report
The manuscript titled: “Microvascular leakage as therapeutic target for ischemia reperfusion injury” raises the important topic of interstitial oedema as one of the key mechanisms of ischemia-reperfusion injury. The article raises a very important issue, there are few studies on this subject. I work with an isolated rat heart myself and have observed interstitial oedema during rapid reperfusion. According to my observations, postconditioning reduces swelling very well. It has been shown in the study of others: https://www.nature.com/articles/nrcardio.2012.94.
I would also cite the article: https://journals.physiology.org/doi/abs/10.1152/ajpheart.00347.2022
The only major objection I have to the “conclusion and outlook” chapter, which concerns the general topic of IRI and there is only one sentence concerning the main theme of the manuscript.
Minor editing of English language required - there are some typos and repetitions (like line 29).
Author Response
Point-by-Point response to the comments of the editor and the reviewers
Cells – SubmissionID: cells-2325034
Microvascular Leakage as Therapeutic Target for Ischemia and Reperfusion Injury
We would like to thank the reviewers for their thorough review of our manuscript and their valuable and helpful comments. We have revised parts of the manuscript and integrated new detailed information for the reader on the basis of the excellent comments.
The reviewers' comments partly overlap thematically. We have tried to incorporate all ideas and comments. Due to the narrative structure, we have had to place some points in several places. To make it easier to find them, you will find the changes highlighted in the manuscript. We are convinced that this has significantly improved the quality of the manuscript. Please find below the point-to-point revision.
Comment 1:
I work with an isolated rat heart myself and have observed interstitial oedema during rapid reperfusion. According to my observations, postconditioning reduces swelling very well. It has been shown in the study of others: https://www.nature.com/articles/nrcardio.2012.94.
Response 1: We have looked at the paper and included it as it contains important points. Thank you for adding your comments to our review.
Comment 2:
I would also cite the article: https://journals.physiology.org/doi/abs/10.1152/ajpheart.00347.2022
Response 2: We happily added the proposed paper.
Comment 3:
The only major objection I have to the “conclusion and outlook” chapter, which concerns the general topic of IRI and there is only one sentence concerning the main theme of the manuscript.
Response 3: Although general IRI is important, we share your opinion on this point. Therefore, we have added further points to underline the specific thematic reference (p14, l.765-772).
Reviewer 3 Report
This review focuses on microvascular damage (MVD), in particular microvascular leakage (MVL) under conditions of IR, which is an interesting topic. The first half of the paper reads good and focused. However, the content in the second part (from page 412) is not what the readership is looking for. In addition, there are some aspects very relevant that need to be included into this themed review paper.
1. The article should emphasize that MVD presents as two main aspects: occlusion and leakage with distinct mechanisms and implications. There has been insufficient appreciation on the latter (i.e. MVL) relative to that of MV occlusion.
2. Findings from cardiac MRI studies on MVD, i.e. oedema (LGE enhancement) and intramural haemorrhage (or hypotensive core) should be summarised with their predictive value pointed out.
3. Trials with negative results on IR protection are not new at all and have been discussed by other review papers. Also, these studies did not generate data on MVD or MVL.
4. There has been increasing number of preclinical studies on Post-IR protection against microvascular damage, including MVL. These studies need to be presented and discussed. Listed are some of them:
PMID: 25932532 on COMP-Ang-1 (cardiac MVD)
PMID: 31218471 on serelaxin or relaxin (cardiac MVD and MVL)
PMID: 36639597 on Imatinib (cardiac MVD)
PMID: 30472160 on Tetrahydrocurcumin (brain MVD)
5. Injury and compensatory changes in Lymphatic vessels have been shown to be important in final cardiac IR injury. The authors might wish to cover this part in this review paper.
Author Response
Point-by-Point response to the comments of the editor and the reviewers
Cells – SubmissionID: cells-2325034
Microvascular Leakage as Therapeutic Target for Ischemia and Reperfusion Injury
We would like to thank the reviewers for their thorough review of our manuscript and their valuable and helpful comments. We have revised parts of the manuscript and integrated new detailed information for the reader on the basis of the excellent comments.
The reviewers' comments partly overlap thematically. We have tried to incorporate all ideas and comments. Due to the narrative structure, we have had to place some points in several places. To make it easier to find them, you will find the changes highlighted in the manuscript. We are convinced that this has significantly improved the quality of the manuscript. Please find below the point-to-point revision.
Comment 1:
The article should emphasize that MVD presents as two main aspects: occlusion and leakage with distinct mechanisms and implications. There has been insufficient appreciation on the latter (i.e. MVL) relative to that of MV occlusion.
Response 1: We would like to thank you very much for this important point. We have conducted a new literature search and expanded the text under 2.1 Myocardial IRI accordingly. This expands the thematic range of the manuscript.
Comment 2:
Findings from cardiac MRI studies on MVD, i.e. oedema (LGE enhancement) and intramural haemorrhage (or hypotensive core) should be summarised with their predictive value pointed out.
Response 2: We have also mentioned this point in the rewritten chapter 2.1.
Comment 3:
Trials with negative results on IR protection are not new at all and have been discussed by other review papers. Also, these studies did not generate data on MVD or MVL
Response 3: We believe it is useful to list the large negative studies to show that clinical research on IRI may have gone in the wrong direction. We then argue that microvascular leakage has been understudied as a pathomechanism and that this could be a new direction. If you do not agree with this idea, we will be happy to discuss it again.
Comment 4:
here has been increasing number of preclinical studies on Post-IR protection against microvascular damage, including MVL. These studies need to be presented and discussed. Listed are some of them:
PMID: 25932532 on COMP-Ang-1 (cardiac MVD)
PMID: 31218471 on serelaxin or relaxin (cardiac MVD and MVL)
PMID: 36639597 on Imatinib (cardiac MVD)
PMID: 30472160 on Tetrahydrocurcumin (brain MVD)
Response 3: Thank you for the proposed papers. We have read them and agree that they are an excellent addition to the review. We therefore integrated and discussed them in our manuscript.
Comment 5:
njury and compensatory changes in Lymphatic vessels have been shown to be important in final cardiac IR injury. The authors might wish to cover this part in this review paper.
Response 3: We have addressed this point in the manuscript under 2.1. As lymphatic events are interesting but would open another big door, we have limited ourselves to a "food for thought" for the reader and provided a source for the passage. We hope that this approach adequately addresses your comment.
Round 2
Reviewer 3 Report
The authors have revised the paper adapting my previous comments. The current version reads good. Well done.
I understand the view of the authors on the section that summarize a number of clinical trials on post-MI cardiac protection. However, it would make this review article better focused and interesting to read if this section could be shortened.
Author Response
- Point-by-Point response to the comments of the editor and the reviewers
Cells – SubmissionID: cells-2325034
Microvascular Leakage as Therapeutic Target for Ischemia and Reperfusion Injury
We would like to thank the reviewers for their thorough review of our manuscript and their valuable and helpful comments. In our eyes, the quality of the review could be increased immensely.
- Point to Point Revision:
Reviewer 3:
Comment 1:
I understand the view of the authors on the section that summarize a number of clinical trials on post-MI cardiac protection. However, it would make this review article better focused and interesting to read if this section could be shortened.
Response 1:
Thank you very much for your comment, we are pleased that we have already been able to substantially improve the review. We agree, that is essential to maintain readability.
That is why, we have shortened and summarised parts of page 10. We have tried not to shorten Reviewer 2's points too much.
We would like to retain what we consider to be important negative studies to show that clinical research on IRI may have gone in the wrong direction. We hope that the review will draw attention to microvascular leakage as a pathomechanism and thus provide food for thought for further research.
We hope that we have found a good compromise between the interests of the readers, the reviewers and our own ideas.